# The Reciprocal Power of Equitable, Intergenerational Learning: Exploring Perspectives of Undergraduate Students about Engaging in a University–Community Partnership Program

John Cano [1,*] and Diana Arya [2]

1   Berkeley School of Education, University of California, Berkeley, Berkeley, CA 94720, USA
2   Gevirtz Graduate School of Education, University of California, Santa Barbara, CA 93106, USA; darya@ucsb.edu
*   Correspondence: jcano@berkeley.edu

**Abstract:** Our qualitative study is a deep exploration into the underexamined notion of engagement with respect to undergraduate students who took up the role of co-researchers within an afterschool program designed to engage young students about environmental issues and sustainability practices. This research program is based on a community-based literacies framework that addresses all members (program leaders, graduate students, undergraduate students, and young students) as co-learners. This study explores the largely unknown experiences of undergraduate students in informal learning contexts, which broadly center youth experiences. We took a critically framed approach (i.e., mindful of institutionalized inequities–known as a kind of silencing—for traditionally marginalized populations) in our analysis of interview responses from 11 undergraduate students involved in an afterschool environmental program for young students living in a Latinx neighborhood in central California. Our analysis involved a two-phase process that began with a general thematic exploration of transcribed interviews followed by in-depth, microlevel transcription of salient instances regarding community engagement. Responses suggest that the community-based context enabled a deep engagement founded on shared cultural practices, life experiences, and engaging in disciplinary projects (e.g., building and planting an edible garden). This study is a contribution to the long-needed insights into the importance of community engagement and leadership experiences during undergraduate learning, particularly for traditionally marginalized students. Findings may be informative to educators and researchers striving to transform college experiences for diverse student populations.

**Keywords:** community engagement; undergraduate students; student engagement; experiences; afterschool; community-based literacies

## 1. Introduction

Over the past couple of decades, there has been a steady increase in afterschool programs, particularly university–community partnerships, to address the call for supplementing educational experiences and career preparation for K-12 students and for supporting community wellbeing (Cole and Consortium 2006; Durlak and Weissberg 2007; Hudson and Hudson 2008; Newman et al. 2000; Tyler-Wood et al. 2012). These sites often foster rich learning for youth, crafting spaces in which interactions are opportunities for building knowledge with others in generative ways (Vygotsky 1978; Durán 1994; Gee 2012). However, the research almost exclusively focuses on youth experience in these programs rather than capturing the broader context of university–community partnerships, particularly the experiences of undergraduates are often the key connector for such endeavors. What remains largely unknown (and, hence, silent) are the ways in which undergraduates are positioned in such programmatic initiatives, and how the ways that they are positioned align with or differ from experiences in higher-education learning.

Assuming a progressive stance on voice and agency as key factors for supporting learning and creative work, we acknowledge and push back on dominant institutionalized learning, which is designed without spaces for undergraduate students to use their voices and exercise their agency to contribute and work for the common good of all participants (undergraduate students, faculty, staff, and youth) of a given partnering community. Without these spaces—with an explicit focus on justice—it would be difficult to avoid colonizing structures (Gordon da Cruz 2017). A few studies have examined relationships between youth and staff, site coordinators, and adult volunteers, to improve quality experiences for youth (Pierce et al. 1999; Mahoney and Stattin 2000; Smith et al. 2009; Hinga and Mahoney 2010), but there is little to no information about the undergraduate students involved in supporting such programs and their experiences. Educational research is largely silent on the perceived roles, responsibilities, and interests of these adults who are tasked to design and facilitate engaging activities for participating youth, particularly within multicultural, multilinguistic contexts. Given the growing number of partnerships between universities and community organizations in the US, it is increasingly important that we understand the undergraduate experience as program facilitators (Delacruz and Guerra 2019; Afterschool Alliance 2014). Hence, our objective in this study was to explore the largely unknown world of undergraduate engagement in university–community partnership programs.

Best practices for fostering rich learning environments have been long described as crafting spaces in which interactions are opportunities for building knowledge with others for culturally relevant purposes (Vygotsky 1978; Durán 1994; Gee 2012). Relevancy plays a large role in fostering students' engagement in educational activities; the more one is engaged in an activity such as composing a digital story about familiar experiences, the more they are actively involved in the learning experience and, hence, tend to have lasting, meaningful impressions of associated content (Munns and Woodward 2006; Guthrie and Wigfield 2000; Fredricks et al. 2004). However, it remains unclear what, exactly, it means to be engaged in an activity or subject matter, as research on this topic seems to undervalue the extent to which the sociocultural context can be unwelcoming to those from nondominant backgrounds. Such research leaves out the contextual realities for marginalized students who are on unequal footing compared with white peers when navigating institutional spaces. The partnering university featured in this study is a Minority-Serving Institution (MSI) and, as such, we were well positioned to fill such a gap in higher-education research.

What does it mean for a culturally and linguistically diverse group of undergraduates to be engaged in a community partnership program? Do high levels of engagement among undergraduate students necessarily involve enjoyment (Shernoff and Vandell 2007)? Collectively, the numerous studies on engagement present an unwieldy picture, revealing little about the range of perspectives of undergraduates who often serve as key facilitators in youth-based programs led by university faculty and staff. While there is some movement toward exploring how undergraduate students view their participation in programs designed for community youth, such research remains few and far between (Arya et al. 2022a).

Given the growing number of partnerships between universities and community organizations, it is increasingly important that we understand the undergraduate experience in such contexts (Delacruz and Guerra 2019; Afterschool Alliance 2014). Without undergraduate support, these programs could not succeed. Further clarification on undergraduate experiences and perspectives may help in sustaining partnerships that could benefit both the university and the surrounding community. Hence, we aimed to learn about the experiences and perspectives of 11 undergraduate students who served as facilitators for a youth-based program called New Leaf (pseudonym), which is an afterschool program involving gardening and DIY activities focused on sustainability and environmental stewardship. Specifically, our study addressed the following research question: What are the perceptions of undergraduate students about their engagement in a community-based afterschool program? We took an ethnographic approach (Mitchell 1984) for this study, which is noted by prominent scholars in survey and measurement practices as an important

first step toward the development of more broadly administered, systematic approaches such as surveys and measures (Wills 2005; Wilson 2005). Our choice of methodology is based on the need for further clarification on theories and constructs most relevant to undergraduate involvement in community learning, thus contributing to the foundation needed for broader systematic explorations. As such, we looked to research on general student engagement as a starting point for our exploration.

## 2. Critically Contextualizing Student Engagement

Our review of research on engagement across the fields of education, psychology, and sociology revealed more confusion than clarity with no mention about how the sociocultural and sociopolitical context can impact a community member's engagement. Many educational studies support the general notion that when students are engaged—broadly described as attentively interested in a particular topic, activity, or phenomenon—they are more involved in the learning experience and in becoming active participants within a given learning context (Munns and Woodward 2006; Fredricks et al. 2004; Arnett 2016; Wentzel 1997). Such a notion is arguably irrefutable, yet in our review of past research on engagement in learning contexts, the focus centered on students as the source of all efforts to engage. Higher-education scholar Trowler (2010), for example, places notions of engagement as hinging on interests in learning about particular topics, such as the history of quilt making. However, what if the presented content privileges some histories such as the quilting traditions of Amish communities over others such as the rich, activistic, quilting roots of Black Americans? It is arguably difficult to be engaged in a topic one naturally finds interesting without the sense of inclusion and representation.

School curriculum scholars Fredricks et al. (2004) took a componential view by suggesting 'engagement' as most effectively evidenced through three key dimensions—behavioral, emotional, and cognitive. While this componential view provides potential differentiated insight on this construct, descriptions across the components overlap in respective domains; a student's behavioral demonstration of engagement by attending a reading club every day that the activity is offered, for example, is evidence of an emotionally charged interest in reading with others. Furthermore, cognitive interest in reading is arguably a prerequisite for such consistent attendance and emotional attachment. Such overlapping of multidimensional aspects can be problematic during research analysis that involves categorizing and analyzing recorded observations, as well as during formative assessment practices. Furthermore, and most importantly, the decontextualized presentations of such dimensions, such as 'behavior' are particularly vulnerable to biases: What behaviors, for example, would best connote engagement? If an educator or researcher sees a child laying their head on a desk, does this mean the child is disengaged? Alternatively, are they thinking deeply about the problem being represented, with their head on the table to block out distractions while engaged in problem solving? Without the explicit encouragement to think ethnographically (i.e., centering on participant values, experiences, and ways of thinking and doing) about such observational phenomena, the viewer has little guidance in self interrogating assumptions about what is happening and why.

Scholars who study engagement within afterschool contexts seem most promising in providing a model for ways of researching engagement, given that most of these studies tend to focus on the community context. For example, Shernoff and Vandell (2007) identified the importance of concentration, interest, and enjoyment during different afterschool activities such as arts and crafts, sports, and completing homework. These researchers compared such activities with what students do during a regular school day by analyzing the levels of engagement observed (i.e., perceived efforts) of students in these respective contexts. The comparative framework followed Flow theory (Nakamura and Csikszentmihalyi 2009; Csikszentmihalyi 1990), which assumes that the level of challenge (difficulty) of a given activity met with a given level of skill of capability determines the level of anxiety, or sense of boredom, a student may experience. When students are in the flow, they are neither anxious or bored, which is presumed to be the ideal condition for learning something

new. The efforts to contrast contextual differences in student experiences brings us closer to understanding engagement beyond what a student is doing. However, what seems unaddressed, or rather 'silent' (Arya 2022) in such work is the role of the undergraduate educator, activity facilitator, or mentor in these community contexts. The bulk of studies about educators or program leaders center on the design of program-related activities or student outcomes that include final project quality and assessment performance (e.g., Hinga and Mahoney 2010; Mahoney and Stattin 2000; Smith et al. 2009; Vandell et al. 2012; Kataoka and Vandell 2013). Can engagement thrive in a community context if those who are leading or facilitating activities are themselves not engaged? This question inspired our present study.

If we accept the assumption that one inspires engagement in others through their own interests and motivations, then it stands to reason that teacher or facilitator engagement is a key phenomenon to unpack when exploring student engagement. Adults in studies about afterschool educational practices seem limited to children's experiences and essential factors for fostering a safe space/environment for youth (e.g., Larson 2000; Eccles and Gootman 2002; Posner and Vandell 1994; Shernoff and Vandell 2008). As such, educational research is largely silent on the sociocultural and sociolinguistic connections, perceived roles, responsibilities, and proclivities of these adults who are tasked to design and facilitate engaging, safe activities for participating youth. Furthermore, there is little understanding about the development of strategies for undergraduate educators and facilitators beyond the process of gaining certification for classroom teaching or for boosting student assessment performance, potentially leaving readers with the implication that the afterschool context is less of an educational priority (Hinga and Mahoney 2010). Simply put, we have little notion about the particular constructs (i.e., ideas and concepts) most relevant to undergraduate engagement in community-based learning, leading to our decision to conduct an ethnographic exploration of a particular community context (New Leaf) as a way of clarifying core concepts that can be employed for future investigations.

Over the past couple of decades, there has been a steady increase in university–community partnerships as a way of supplementing educational experiences and career preparation for K-12 students and supporting community wellbeing (Cole and Consortium 2006; Durlak and Weissberg 2007; Hudson and Hudson 2008; Newman et al. 2000; Tyler-Wood et al. 2012). Given the influx of adults working to enhance the engagement of youth beyond classroom walls, a deeper exploration into the inspirational origins of such educators and facilitators of engagement in learning is warranted. The observed lack of knowledge about adult engagement is coupled with the lack of clarity about what constitutes engagement, leaving us with the challenge of finding an adequate approach for exploring engagement in our current study. As such, we took up critically framed epistemological approaches that together offer a contextual anchor for exploring notions of engagement in a community–university partnership program from the perspectives of undergraduate students who serve as undergraduate facilitators and near-peer mentors for participating youth. Specifically, we used critical community engaged scholarship (Gordon da Cruz 2017), as well as the notion of silence in communicative contexts (Arya 2022) for the theoretical groundwork in our study, which is guided by the following research question: How do undergraduate program facilitators view engagement within an environmental, community-based program for youth?

### 3. Theoretical Framework

**Critical Community Engagement Scholarship.** In order to successfully connect with communities, scholars must be committed to engage with community members in meaningful, collective, and culturally responsive ways. All members within a community are knowledgeable and experienced stakeholders regardless of their age. Such scholarly efforts in connecting and working with community members as equitable partners aligns with the core principles of the framework known as Critical Community Engagement Scholarship (CCES) introduced by Gordon da Cruz (2017). Gordon da Cruz described six principles

key for engaging with communities, and we found this work to be particularly helpful for how we listened to our undergraduate team of co-researchers and what they thought about learning, working, creating, and playing alongside their younger co-learners. New Leaf is one of six community-based programs that constitute the multiprogram initiative called Community-Based Literacies (CBLs). All CBL programs are founded on three core values—agency, co-learning, and belonging (Arya 2022). As such, we found Gordon da Cruz' framework particularly relevant for our study about undergraduate experiences in the CBL program New Leaf.

The CCES principle of defining societal issues with community connotes the importance of fostering a sense of agency among our community members who take part in the decision-making process about the kinds of goals and activities we pursue in our partnership. Another principle highlights the importance of involving community members as research colleagues who engage in scholarly investigations with university team members. The co-learning nature of CBL programming necessitates the practice of viewing and positioning all members—youth, undergraduates, graduate students, staff, parents, and faculty—as equal members of a scholarly endeavor. Hence, our community-based program is fully collaborative and mutually beneficial to all members as described with the third CCES principle. The knowledge and creative work we produce in this collective directly addresses public interests and/or issues (the fourth CCES principle); New Leaf activities and projects center what is happening, what is missing or wrong, and what we can do together. One project involved co-learners contemplating, brainstorming, and transforming an alleyway next to the youth club into an edible garden. This vivero soon became a treasured space for the club and the surrounding neighborhood; rather than rushing home to prepare dinner, parents lingered to chat and pick up any ready fruits and veggies to add to their meals. Resources from the university supported such community interests (fifth CCES principle); the young co-learners teamed up with the undergraduate co-learners to create digital diaries—with iPads provided by the University—where they could keep track of the health and growth of their plants.

Such digital activities afforded by institutional resources hinged on the community-driven goal of cultivating a garden. This example also represented the planning that began in a community-based practicum course through which the study participants selected to support New Leaf. One of the lead instructors of this course also leads CBL programming that includes New Leaf. This faculty member (second author) explicitly integrates such teaching and programming with their research (sixth CCES principle), which focuses on a number of sociocultural issues related to literacies, multilingualism, and innovative practices and assessments designed to include community interests and goals (e.g., Arya n.d., 2022; Arya et al. 2022a, 2022b; Karimi et al. 2023).

The alignment between the CCES principles and the CBL program context is evidence of the appropriateness of using this framework as a theoretical guide for our study. While the principles can help organize and clarify interpretations of participant responses, there may be unspoken matter that may also warrant exploration.

**Exploring the 'Void' of Institutionalized Learning.** Context matters, and much of THE undergraduate experience is shaped by rules, attitudes, and practices that are unspoken and unwritten. In a previous study on undergraduate experiences (Arya et al. 2022b), we found in our analysis of interview data a number of invisible factors that inhibit a sense of belonging and inclusion, even in a university with a culturally and linguistically diverse student population. The participants' engagement in CBL programming such as New Leaf revealed the ways in which higher education institutions inhibit the inclusion of interests and aspirations that are explicitly supported in CBL. Hidden systemic processes and practices can be viewed as a kind of void, one that is noted by scientists as 'the known unknown' (Arya 2022). We learned from our interview conversations with participating undergraduates a number of debilitating institutional inequities, including the possibility that institutional resources may not be viewed and utilized as intended. Participants with Latinx roots noted a reluctance to attend faculty office hours in order to avoid being viewed

as lacking intellectual abilities compared with their white peers (Arya et al. 2022b). Such a discovery prompted the reinvention of this resource as a required assignment for all students enrolled in a given course.

By positioning our participants as cultural guides, we are engaged in learning what may not be visible from a researcher stance, thus warranting efforts to engage in ongoing exchanges throughout the study to clarify our interpretations. Acknowledging the void means that we are willing to acknowledge that many influences exist within a sociocultural context, including racism, sexism, and power dynamics, all of which impact a community.

Using the CCES framework with a mindful eye on the institutional void of undergraduate experiences, we aim to clarify what it means to be engaged in a community-based program for undergraduate students.

## 4. Methodology

### 4.1. Study Context

Our study took place within the context of an afterschool club called New Leaf (pseudonym) which is a university–community partnership designed to foster multilingual and multimodal (story writing, video and podcast creations, etc.) literacy-related activities related to local environmental topics and issues. In this New Leaf program, undergraduate students learn alongside their young co-learners (grades 5–8) about farming, gardening, and critical observations about the environment (i.e., how to be aware of our surroundings, and how to observe and protect local natural surroundings). New Leaf sessions occurred twice a week; one of the weekly sessions took place at the club site within the vivero while the other took place at a previously determined location on our university campus (e.g., campus greenhouse).

New Leaf reflected a community-based approach grounded in three values that were explicitly discussed during biweekly sessions—agency (participants positioned to take leadership roles and contributing ideas, knowledge, and expertise to activities and projects; co-learning (all members young and old are positioned as knowledgeable others with unique and valuable experiences and skill sets); belonging (all members are unconditionally valued members who are able to see the program as a welcoming space for collaboration). Creative work produced from co-learners included poetry, open-space (nature) explorations, researching digital sources, writing, gardening, art activities (painting, crafting, and drawing) and technology-related activities (photographing, videotaping, creating, blogging, and producing digital narratives/storytelling); for example, we brought iPads to our vivero space so that young and undergraduate co-learners could track the growth and overall health and 'happiness' of plants.

During the time of this study, our undergraduate co-learners, 11 in total, had just ended the spring term programming, and all local schools (including our university) were in the final days of the academic year. All 11 undergraduates agreed to participate in this study.

### 4.2. Participants

Our study includes the recorded interview responses of 11 undergraduate students who are pursuing a minor in education at our research university. All participants served as facilitators in the afterschool program New Leaf and had supported this program for at least one of the three academic quarters (10–25 weeks). These facilitators were recruited through an education minor course that is taught every term throughout the year. This course provides the students a set of different site placements to volunteer as facilitators; each site has a particular goal (e.g., assisting these undergraduate minors in the beginning pre-service phases in becoming classroom teachers)

Out of the 11 participants, eight (73%) identified as women, and three (27%) identified as men. Four (36%) of them spoke only English, and seven (64%) were multilingual (five spoke Spanish; one spoke Japanese; one spoke Farsi). Five of the participants (45%) identified as having Latinx roots, four (37%) identified as white, one (9%) identified as

Middle Eastern, and one (9%) identified as a mix of racial groups (Native American/white). All participating undergraduates were taking upper division courses and were in the final (senior) year of their studies.

### 4.3. Epistemological Framework

Our approach to this study follows Mitchell's (1984) notion of a 'telling case' This ethnographic study is a 'telling case' (Mitchell 1984), which is understood as a rich, multi-layered study of a particular group of individuals who are working together to accomplish a particular task (or set of tasks) for a particular purpose and within a particular sociocultural context. Given the context of our study, which centers on the context of collaborative projects and activities associated with New Leaf, we found this telling case approach to be appropriate. The rationale for such a context-specific community approach aligns with educational research scholars who have long emphasized the importance of deep qualitative explorations into little known phenomena prior to investigating broader contexts (Wills 2005; Wilson 2005). Such pointed explorations, according to these scholars of survey and measurement design, contribute to the theoretical foundations that can strengthen broader subsequent studies.

All notes, interview questions, and other associated data sources for our study aligned with the overall purpose of gaining an 'insider's perspective' of experiences and perspectives of our participants. Hence, we constructed questions with the singular purpose of understanding the notion of engagement from our participants' perspectives, considering the need to be mindful of our respective positions and potential biases in interpreting responses.

### 4.4. Positionality

The first author of this study led the process of interviewing participants who had connected with him on a regular, weekly basis prior to the study. As program coordinator, the lead author took the time to learn about the interests and aspirations of the undergraduate students who would (based on field note exchanges) often seek support and guidance about potential activities with youth. Like all members of New Leaf, the coordinator positioned himself as a co-learner and, as such, fostered a sense of equity during planning meetings and program sessions. He also brought his knowledge and expertise in technological applications when given the opportunity.

The second author of the study provided consultation support as needed for both program-related activities and research efforts. As program leader, they were responsible for all reporting obligations to various funding agencies, as well as all creative work produced from programmatic activities. Similar to the first author, this leader explicitly positioned themself as a co-learner, being mindful of the potential power dynamics during exchanges with New Leaf members. Hence, we both engaged in discussions with young and undergraduate co-learners with an explicit goal of positioning younger co-learners as knowledgeable others and central decision makers. This goal also served as a compass for data gathering and analysis for this study. Furthermore, we aimed to be mindful of potential cultural and linguistic biases in our efforts to interpret interview responses. As such, we maintained a mindful connection to our respective identities (first author as Latinx multilingual and second author as Southwest Asian North African/white biracial multilingual) in order to more effectively unpack the knowledge and experiences of our undergraduate participants.

### 4.5. Data Sources

Ethnographically guided interviews. Participants took part in individual conversational interviews (Skukauskaitė 2017), which were guided by an overarching question: What does it mean to be engaged as a facilitator in this community-based afterschool program for youth? Such an approach to questioning is the ethnographic alternative to more systematically designed interview protocols; questions are loosely structured to represent a

natural conversation. The purpose of such an approach is to ensure our ability to capture what our participants viewed as most pertinent to their engagement (Mitchell 1984).

These interviews reflected the structure of natural conversations that were loosely guided by the following items: Tell me about your experiences in this program, what it's been like for you. What does it mean to be engaged as an undergraduate in this program? How is such engagement similar to or different from your expectations before coming to this program? How did you feel about our encouragement to take a leadership role in developing an activity or project for our program? What are you most proud or happy about creating for this program? How much do you see yourself connected with this program? How connected do you feel with all the members of this program, both young and older co-learners? How do you think this community program 'fits' within your undergraduate program goals? What activities or projects were most meaningful to you?

Taking a conversational style meant that these questions were not verbatim, and adjustments aligned with information shared in responses to previous questions. Such a conversational style allowed for a more responsive approach to each participant, which has both shared and unique experiences and associations with the New Leaf program. All the interviews were audio-recorded and transcribed at a large-grain level; only uttered words and pauses (indicated by periods) were captured prior to analysis.

**Written member-checking responses.** One-page summaries of initial findings from interviews were sent to all participants via email in order to provide an opportunity to review and clarify our interpretations. While follow-up discussions on this member-checking summary were not required, five participants responded to the invitation to share more or clarify points raised in the summary (see Appendix A).

**Field notes.** The first author served as the program coordinator during this period, thus providing opportunities for capturing salient moments mentioned by participants in situ. Such localized knowledge fostered a sense of trust and connection with the undergraduate participants. Notes were kept digitally through exchanges with participants and program leadership (second author) and, as such, spanned 40 exchanges across the duration of the study.

### 4.6. Analytic Framework

Our analysis centers on the 11 audio-recorded, roughly transcribed interviews, which followed a four-phase process. The first phase involved creating macrolevel summary statements of participant utterances in order to identify key points about programmatic experiences and perspectives. Such statements reflected perspectives on programmatic values (agency, co-learning, and belonging) that also align with key CCES principles. The second phase involved a deeper exploration of responses deemed most salient for our study. To be clear, our two-phase exploration is a kind of grounded approach to understanding what qualities, practices, and experiences were most valuable to our participants. As such, we were not clarifying variables as might be more applicable to traditional research studies. As an ethnographic exploration, our efforts were focused on the language and constructs most salient for our participants.

We transcribed these salient instances into message units, which constitute the minimum representation of a speech and allow to make meaning of the message shared by someone (Bloome et al. 2004). These message units support our goal to gain an insider's perspective as previously mentioned; essentially, we aimed to represent how the participants constructed and built their responses to our questions, thus allowing us to closely listen to and (re)present our participants. Throughout these microlevel transcriptions, meaningful contextualized cues, such as an increase in volume (through use of bolding) and changes in intonation (arrows at the end of a given unit), were highlighted. The third phase of our analysis involved a thematic interpretation of message units using the aforementioned CCES aligned lens (Gordon da Cruz 2017). Specifically, we interpreted message units by connecting implicative words and phrases with programmatic values (agency, co-learning, and belonging). Below is an excerpted example of such fine-grained analysis of one of

the participants, *K* (initials of participants' names were used as pseudonyms); the circled portions within each message unit below connoted respective principles indicated on the left (see Figure 1 below).

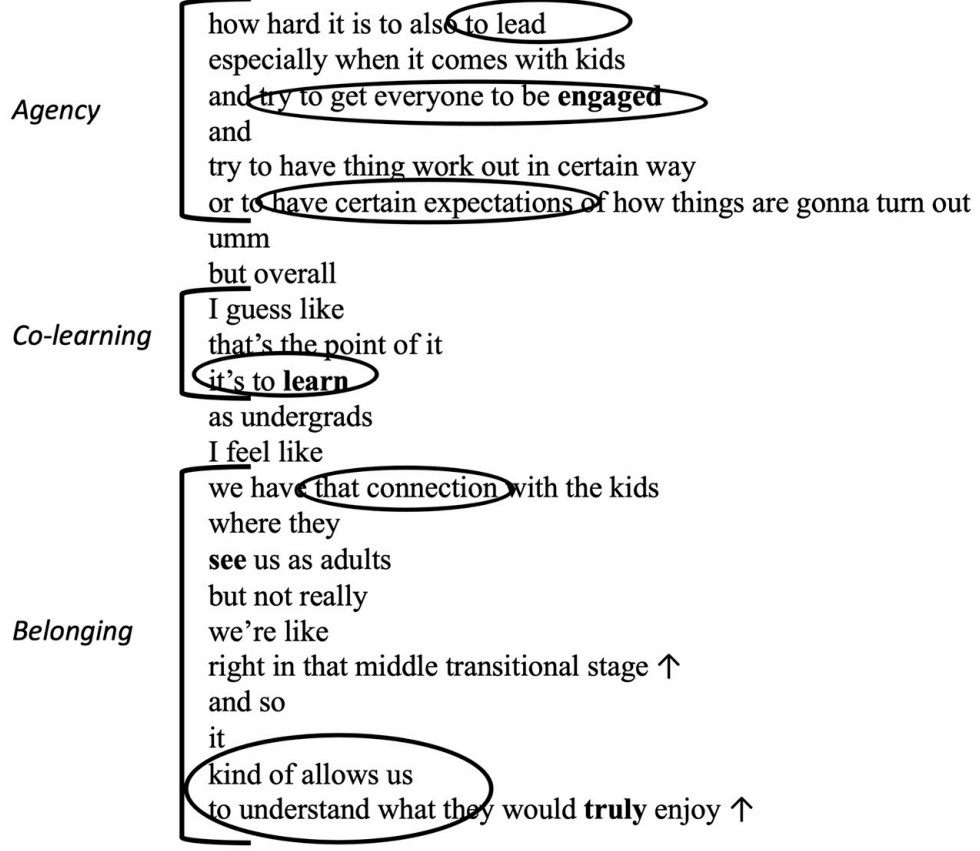

**Figure 1.** Excerpted analysis of message units.

From such programmatically organized message units, we proceeded to the fourth and final stage of our analysis, which involved a community check of understanding with the participants about our interpretation of recorded conversation. was done in order to get a more accurate representation of their perceptions of engagement in the program. Excerpted responses to the member-checking summary were included along with our interpretations and follow-up commentary from five participants (see Appendix A). None of the participants pushed back on summarized findings; all commentary reflected a desire to emphasize key points. Contextual affordances of gathered field notes helped in clarifying participant commentary on particular program activities as needed.

## 5. Findings

What does it mean for undergraduate students to be engaged in a university–community partnership program? All interviews revealed an evenly distributed connection to each of the three programmatic values (agency, co-learning, and belonging). Our findings from analysis of interview questions and subsequent member-checking commentaries suggest that engagement within the context of the New Leaf program is an interactive network of key qualities that participants believed were core to how they were engaged in the New Leaf program. Figure 2 shows an overlay of these qualities on programmatic actors within this program.

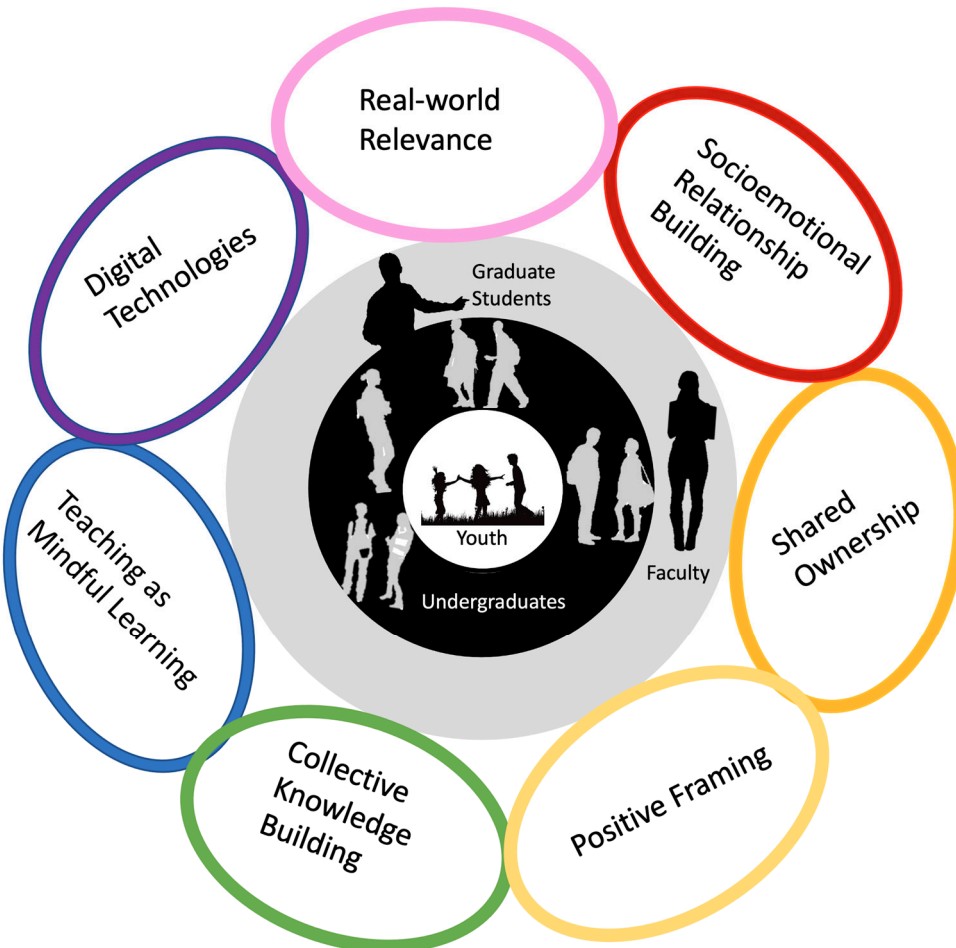

**Figure 2.** Reported key qualities of engagement in New Leaf.

As described earlier, members in New Leaf are concentrically organized on the basis of their respective positions and roles in the program, establishing the concentric configuration in the above illustration. All of the participants agreed with the presented interpreted qualities of engagement (see Appendix A) that are reflected in the surrounding linked rings above, which are deliberately connected given the observed overlap in how participants expressed their thinking. According to the undergraduates, engagement reflects a democratic, mindful effort of acceptance and acknowledgement of knowledge and expertise of all others, young and old. The undergraduates agreed that such engagement should also reflect the digital era that we live in, and expand the number of connections and ways of communicating with one another. Each of these seven qualities of engagement is described in turn.

**Real-world relevance.** Participants consistently highlighted the importance of hands-on, authentically anchored projects as a way of engaging members to work together. Participant EB (Chicanx/Latinx male) highlighted the value of the place-based nature of learning in his comment about which experiences were most engaging:

> Being able to spray the **water**
>
> throw the **seed**
>
> throw whatever was on there
>
> to learn how
>
> you know
>
> their local watershed
>
> to learn what's happening in the oceans around them



> I think that's the [activity] where they learned a lot about what happens
>
> right in their **own** backyard

Participant TH (Middle Eastern, female) expressed a sense of awe about the real-life connections between learning and creating:

> Like the garden is there to grow plants
>
> and so we grew plants
>
> and so
>
> that was like
>
> the **core** activity
>
> and also like
>
> we're all learning and working together
>
> **very** hands on
>
> so I think that was really great
>
> they're **eating** the salad
>
> we like
>
> **grew** these
>
> like
>
> from nothing
>
> and I thought that was **really** really really incredible
>
> and like
>
> an opportunity
>
> they might
>
> they
>
> probably would never have otherwise

The sentiments represented in the above excerpted quotes reflect an appreciation for and the rarity of such opportunities to learn and collaborate in a way that engages one with the world around them.

**Socioemotional relationship building.** All participants emphasized the importance of taking the time needed to make meaningful connections with others in the program. For example, Participant EE (white female) emphasized the importance of bonding with her peers, serving as a kind of foundation for engaging with youth:

> I think that one thing that I
>
> I guess
>
> not that I would change
>
> but that I've learned
>
> is just the importance
>
> between facilitators
>
> for us to like
>
> **bond** with one another
>
> and get to know each other
>
> just because
>
> if our team is strong ↑
>
> then like
>
> it is going to be easier
>
> to go and work with kids

Participant EB continued to share what it was like engaging with youth:

I felt like

one the students

her and I have **really** connected

with each other

since the beginning

and she was like

everyone was like

oh you know

it is going to be **hard**

but everytime I **see** her ↑

we sit next to each other

we **talk** to each other

so I feel like

I've been able to really

connect with her ↑

and I don't know

if it is that

we both wear **glasses**

or we both

speak **Spanish** to each other

The engagement that Participant EB experienced with youth spanned various shared qualities that he seemed to associate with the ability to build a bond with a youth participant. Participant KA (Chicana/Latinx female) seemed to echo this sentiment with further explanation:

Most

of the kids

have also some type of

Latino background

**that** also helps a little bit

because with that comes

like

some cultural aspects to it

that you also **understand**

you know

the fact that we can

relate to a lot of things they are

going through

it really helps

**Shared ownership.** Another consistent sentiment was the importance of engaging in activities and projects as key agents of activities that lead to tangible creations that are owned by the creators. Participant TH describes such ownership from collective engagement in the vivero:

I don't think people get

excited about eating salad ↑

[laughter]

because salad is like

healthy and boring ↑

but people were really excited

to eat the salad ↑

because we **grew it**

and then

again

there is like **ownership** ↑

with the salad

because

**we** made it happen

and we were there

from **start** to finish

For Participant KA, such co-ownership seemed to assuage potential tensions during brainstorming sessions:

Having the agency

to not only plan

our own activities ↑

but also

you know

take in opinions

and

also critique

other people's activities

without having anyone feel **hurt**

or disrespected

at no point

at least for me personally

have ever felt like

there's been like

a kind of hierarchy

you know

like I've never felt that

umm

so I think that's because you guys have also created an atmosphere

at least for the undergrads

where we're more than welcome

to express whatever we're feeling

whether we feel like it was a good day or a bad day

umm

so in that sense

I think you guys [graduate students] have **succeeded** in making us feel like

we're just as part of this community program

as you guys are

On the basis of the participant responses exemplified above, shared ownership seemed to support an openness to engage in new ideas from other members that leads to creations upon which all members can see their mark.

**Positive framing.** All participants agreed that a core quality of engagement in New Leaf was the active, explicit effort to engage with a positive mindset, particularly with youth. Participant TH explained the importance of such a mindset:

The [project] may come and go

but I think like your personality

and just like

that constant memory of

how someone interacted with you

is like

longer-lasting than just like

just the [project] ↑

so

even though that's like more

substantial ↑

and you can like

hold it

like physically

like

I think that the personality is like

more

what they'd remember

like that **positivity** in the space

Participant MM (Chicana/Latinx female) emphasized the importance of engaging with youth as friends rather than authority figures:

I think

it was very **valuable**

for

the students

the kids to have

older people

that they can still talk to ↑

without the intimidation

or the **fear** of the **authority**

and

being able to be friends with

people are not your age

whether it's us as facilitators

or faculty

Hence, positive engagement seemed particularly important for our participants in terms of how they connected with youth who may forget the particular activities accomplished, but will always remember how they felt when connecting with college students.

**Collective knowledge building.** A consistent reaction captured across all recorded exchanges was the awe in discovering that one can learn from anyone, regardless of age. Participant MM demonstrates this sentiment in the following excerpt:

It wasn't just us

teaching the kids

information

it was like

we were both **learning** together

there is a lot

that **I** learned

from

the sessions as well

and

it was fun to be like

oh

did you know that before ↑

I didn't know that **either**

and just

umm

learning

it was a whole two-way street

like they would sometimes

teaching me things

that I didn't know about

For participants, engagement in knowledge building involves the act of discovering or creating something that serves as a learning experience for all members.

**Teaching is mindful learning.** Connected with the engagement of collective knowledge building is the positionality of the instructional agent, who were our undergraduates, brainstorming, planning, and implementing activities to engage their young co-learners. All participants agreed that one of the best ways to engage in programmatic sessions was simply to be open to learning. Participant EB explains such active engagement in learning:

When I did the compost bin

I never drilled

I was like

oh you know

I've seen people do it

but right away

[young participant]

was like

I know how to drill

and I was like

I'm gonna trust them

I am going to trust [young participant]

and I was like

you know

show me how to do it

and right away he started drilling

and

he taught other kids how to do it

he taught me

and other facilitators

Participant FB (Chicanx/Latinx male) seemed to suggest that, as described about knowledge building, learning can come from all members:

I think I learned from the kids

but also from

like

the faculty

and in terms of gardening

[undergraduate] knows a lot

so I am

learning from him as well

Similar experiences resonated with field notes from the graduate coordinator who shared his fascination with the experience of learning how to use a sewing machine from a youth participant who led the creation of outdoor seat cushions from recycled materials. The coordinator also took note of such moments of learning across the layers of actors outlined in Figure 2 above, whereby, during debriefing sessions, undergraduate students would comment on what they gained from a previous session and how they may proceed the following week.

**Digital technologies.** All participants agreed that digital technologies were a key part of their activities with youth, and they served as important anchors for communicating thoughts and ideas with others. Participant TH explained the usefulness of the iPads that the graduate coordinator would bring to every session:

I think what I liked a lot

about the iPads

is that

like

you would **comment**

like you put a **comment**

I think **that** was

very interesting to me ↑

to see like

they put like emojis

on each other's

like

posts and things like that ↑

so like

the fact that they can support

each other ↑

and I think like

as someone receiving those comments

on your plant

like that's exciting ↑

so I think that was part

of the **fun** of them using the

iPads

so I think it **added**

to their connectedness

with

their plants and each other

The program coordinator highlighted options for incorporating technologies into weekly sessions and provided training as needed. All undergraduates expressed that they viewed such expertise as an effective way of engaging youth. Participant FB also shared his surprise about the ways in which technologies were integral with other hands-on projects:

It was interesting

merging the technology with the gardening thing

'cause I feel like the QR codes

the iPads

the cameras

everything was kind of **fluid**

I didn't think it would work that well

but it did

and that was surprising

and then **obviously** I was seeing the

you know

I guess

the labor of their work like right there in front of my eyes

Based on participant commentaries, digital technologies played a key role in both documenting and communicating about creative work accomplished during programmatic sessions. Hence, engagement also involves the ability to connect and reflect within asynchronous spaces.

## 6. Discussion

The purpose of this study was to explore what it means to engage in a university–community partnership for undergraduates. It is important to acknowledge that contextual factors matter when exploring the topic of engagement and as such, we made explicit efforts to share with readers the particular programmatic goals, positions, and general environmental qualities of the partnership program called New Leaf. It is also important to acknowledge that, for other partnership programs, engagement may take a different shape and emphasize different qualities. The particular values of New Leaf programming—agency, co-learning, and belonging—present a relevance with the principles of community-based, critically engaged scholarship as outlined by Gordon da Cruz (2017), thus suggesting a particular relevance for this special issue. Simply put, we approached this study with a mindful eye on centering undergraduate students' voice and agency and, hence, pushing back on dominant institutionalized learning practices, which represent a hegemonic, hierarchical structure that often silences those from whom we need to learn in order to understand what is happening, and what is being accomplished within a collaborative space (Arya 2022).

**Key Qualities of Engagement**. On the basis of responses from our undergraduate co-learners who supported the New Leaf program, to be engaged in this program is to be willing to learn from others, including peers and young co-learners, about topics

and issues that have real-world relevance to the community being supported. To be engaged in this program means that academic learning is not the only goal; first and foremost, undergraduates discovered the importance of building understanding and trust with their young co-learners. It is from such trust that all members can feel a sense of belonging and ownership during collaborative, creative work. The questions we posed to our undergraduate participants were designed to unpack perspectives and experiences that may not have been visible to us; both the style of our interviews (conversational) and explicit centering on participant experiences seemed effective in eliciting lived moments of how undergraduates learned about the importance of approaching community youth with positivity. A young co-learner sitting by themselves away from a group activity, for example, was an opportunity to connect rather than reprimand. We also gained insight into the awe that undergraduates experienced about how all members can learn something new from a collaborative project—building skills (e.g., drilling), the growing process of particular plants, the filtration function of our local watershed, and the like—and how such knowledge can come from any member of the community, including youth. Our participants viewed teaching as involving learning, and that being a teacher may be best characterized as a co-learner that also supports the learning of all within a group.

**Knowledge building in the digital era.** Digital technologies (i.e., the use of iPads and associated recording and communicative applications) were viewed by all participants as important tools for fostering engagement in New Leaf. The contextual factor likely most impactful on this shared viewpoint is the technological expertise of the program coordinator who had a background in computer science and positioned himself as an avid explorer of technological innovations for educational purposes. In addition to the aforementioned digital journaling, this coordinator provided resources and training on a number of ways of integrating digital technologies and applications to the program. Resources included mobile apps such as Seesaw as a way for members to share their work and provide feedback/comments to each other, filming and photographing activities and various produced artifacts using iPads (e.g., best usage of lighting and angles with the iPads), and video-editing tools used for rendering video footage collected throughout the year (using iMovie). Co-learners used the iPads to create a timelapse showing the transformation of the vivero over time. These iPads were also a way to create a poetic performance on the recent mudslides nearby, and how such environmental disasters impact the most vulnerable communities. It seems that the knowledge and expertise of program leadership may shape the nature of engagement for a particular program.

Our analysis of recorded conversations along with follow-up member checking with participants revealed seven key qualities of engagement featured in Figure 2—real-world relevance, socioemotional relationship building, shared ownership, positive framing, collective knowledge building, teaching as learning, teaching as mindful learning, and digital technologies. The process of member checking was particularly useful for identifying such prominent features; the confirmation of and additional information to our summarized interpretations helped in clarifying participant views. While member checking is not often used due to feasibility or other factors, researchers may discover more about their participants when checking in with their participants who are important cultural guides into phenomena of interest (Arya et al. 2022a).

*Implications for Community-Based Research*

**The importance of exploring the contextual ground.** Understanding what it means to be engaged in educational contexts, as demonstrated by this study, may be best viewed as a contextualized, ethnographic exploration, which is not an easy task. The findings of this study support an approach where two important variables need to be understood in order to define or understand what engagement means—the context and the participants involved in this context. What made this afterschool program context different from contexts described in other studies about engagement (e.g., Fredricks et al. 2004; Trowler 2010; Csikszentmihalyi 1990) was its community-based approach, where all participants' ideas,

knowledge, and skills were important and valued. Such a community-driven program shaped the expectations of participants who were encouraged to see every interaction in the program as a learning opportunity in multiple ways and directions among the intergenerational community of co-learners. Notions of 'lacking' knowledge or expertise was viewed as an opportunity to learn. Previously mentioned studies on engagement that positioned this construct as a universal, context-free quality of learning miss the importance of the sociocultural context and the often-hidden expectations that shape it.

**Eliciting insider perspectives in community research.** Researchers also would do well to take an insider perspective on programmatic research whenever possible. For our study, we could think of no one more suitable than our participating undergraduates to help us clarify what was important for them in terms of their engagement in New Leaf. Gaining an insider perspective involved a series of data gathering and analytic phases, all of which hinged on positioning the undergraduates as our cultural guides. Hence, our participants were not 'subjects' to be viewed through summative tests and surveys. They were willing collaborators in this process of knowledge building. What is most telling for us about the democratic nature of our programming is the fact that *all* undergraduates that were involved in this study expressed positive sentiments to participating in this study. As one participant (FB) stated to the faculty program leader, "You see us as your colleagues, and I know that you want our voices to be heard". The ways in which we interacted with undergraduates and youth in this program established a foundation of trust that in turn strengthened our ability to explore experiences and viewpoints. Such trust enabled a shared agreement that we would make all efforts to avoid presumptions and biases about participatory engagement.

**Implications for future studies.** Our study may be useful for educators and educational researchers involved in similar community-based work and interested in learning more about the experiences of members engaged in community-based programming. Rather than thinking of engagement as a preset series of values, we think that our study shows how a research team can build a set of conversational prompts or questions on the basis of the explicit goals and norms of their program to learn how various members (undergraduates, youth, families, etc.) think about what it means to be engaged within such a context. For example, university faculty may find it helpful to have more open-ended discussions with their university teams (including undergraduates) at the beginning of programs in order to foster collaborative planning and collective leadership practices. Such efforts may be helpful in reimagining program goals and practices for fostering greater engagement. Our checked interpretations of undergraduate views suggest that our programmatic goals seem to align with experiences of undergraduates who, in a related study on higher-education experiences (Arya et al. 2022b), shared a number of negative experiences and perspectives about studying at a research university. The example mentioned earlier in the introduction about the avoidance of 'office hours' (based on a fear of being viewed as incompetent) was one of several noted findings from this study; participants shared their concerns about feeling 'pushed out' of their original major due to the common practice of having 'weed out' courses. One participant, EE, commented that her experience in New Leaf "helped me see what I don't have as a student at this university". As such, community-based research may be useful in highlighting silent inequities in higher education institutions.

The findings in this study bring up the importance of engaging undergraduate students as integral colleagues within the program, elevating their ideas and allowing them to take leadership roles in the community (whether it is making a tangible activity for the program or supporting and commenting on others' ideas). The perceptions of these undergraduate facilitators make visible how important active participation and interactions are in order to model an active learning among an intergenerational community. Our findings also signify the benefits of researching with rather than about a particular population. Ethnographic work as represented in this study can be one way to engage in research about participant

perspectives, thus providing a clearer view on what is actually happening and what is important according to those we wish to research.

It is also important to note the importance of fostering a flexible community-based curriculum that is adaptable to the needs and interests of all members within a community program. While we emphasized the goals of agency, co-learning, and belonging; we did not dictate how such qualities should look and sound like. Such flexible approaches may help to foster meaningful experiences (engagement) in afterschool programs, supporting and complementing approaches brought by other scholars (e.g., Hinga and Mahoney 2010; Larson 2000; Eccles and Gootman 2002; Posner and Vandell 1994; Shernoff and Vandell 2008; Pierce et al. 1999; Mahoney and Stattin 2000; Smith et al. 2009; Cano et al. 2021) who have studied what factors improve the quality of afterschool programs from different perspectives (e.g., youth experiences, curriculum, and staff and educator quality), especially when there are undergraduate students involved as educators or facilitators. The expressed perceptions and experiences of our undergraduate participants may help others striving to enhance engagement within a particular learning community.

## 7. Coda

We wish to note that the activities and experiences explored for this present study reflect a time just prior to the onset of the COVID-19 pandemic. The New Leaf program was paused for a few weeks while our university team banded together to recreate the kind of engagement that was possible in person. We again embraced the opportunity to learn and create something new together. Soon after a few virtual sessions (via Zoom) with youth, we found new ways to expand connections across cities, states, and countries. We hosted our first Youth Summit that involved young co-learners from Seoul, Korea, and Costa Rica. Throughout this surreal moment in history, we maintained our positive dispositions and willingness to learn. The coordinator noted how "really engaged and talkative" the young co-learners were, and how it was "awesome to see such a multicultural environment happening through zoom, where everyone seemed to be engaged and into it" (based on field note exchanges). While we did not interview undergraduates during this time, we did observe the same kind of engagement captured in this study. As such, if there is any lesson to be learned from our community-based, critically engaged work, it is that, if a research team holds fast to their values, they can overcome obstacles, and perhaps learn something new about the world and each other. In future studies, we may return to this belief in holding onto values, and possibly explore the tensions between the importance of staying true to who we are, as well as the importance of allowing new ideas and shared values to enter the social space.

**Author Contributions:** Conceptualization, J.C. and D.A.; methodology, J.C. and D.A.; validation, J.C., and D.A.; formal analysis, J.C.; data curation, D.A.; writing-original draft preparation, J.C.; writing-review and editing, J.C. and D.A.; supervision, D.A.; project administration, D.A.; funding acquisition, D.A. All authors have read and agreed to the published version of the manuscript.

**Funding:** This research was funded by University-Community Links. The views and perspectives expressed in this publication are solely those of the authors.

**Institutional Review Board Statement:** This study was approved by the University of California, Santa Barbara's Office of Research (#5-22-0578).

**Informed Consent Statement:** All the participants consented to participate in this study.

**Data Availability Statement:** Anonymous transcripts can be provided upon request.

**Conflicts of Interest:** The authors declare no conflict of interest.

## Appendix A. Member-Checking Summaries with Member-Checking Input

| Summarized Interpretations from Interview Exchanges | Additional Input from Participants (a slash—"/"—is used to demarcate summarized input from different participants) |
|---|---|
| Learning should be relevant to one's life, contextualized as phases of learning lead to a real-world product or purpose. | Such relevancy should reflect interests and goals of youth, which are in turn activitely supported by undergraduates. |
| Learning is more than academic; it is a socioemotional endeavor that includes relationship building. | It is also important to bond and create strong connections with other [undergraduate] facilitators in order to strengthen group collaboration. / There is joy in co-creating something bigger than what one can do alone. / Knowledge and expertise of faculty and graduate students did not stifle youth who freely shared their thoughts. |
| Community members must feel some ownership in learning goals. | Faculty and graduate student coordinators would do well to trust undergraduates in contributing to and leading activities and projects. / Teachers benefit from the opportunity to mentor others who are newer to a learning community. / Undergraduates played an important, unique role in the program as near peers, who are positioned to connect with youth that isn't as easy for older adults. / All members (particularly youth) should have ample opportunity to share ideas and opinions so that all voices are represented in planning and collaboration. / Such co-ownership encouraged creative thinking when planning activities and projects. |
| Positive framing is important for fostering learning and engagement in collaboration. | Celebrating **shared** knowledge and cultural roots can be an effective way of engaging students who seem distant during a particular activity. / Expressing good energy and attitudes can leave a memorable mark on youth. / I |
| Students have important cultural, linguistic and experiential knowledge and skills that enhance collaborative work. | Such knowledge must be acknowledged not only for youth but also undergraduates; sharing and modeling skill sets can be motivating for youth. |
| Teachers must be present and mindful learners in order to be effective. | Co-learning can happen among all members of a community (graduate students, faculty and facilitators) and thus not only limited to the youth. Young students have much to teach adults. / Teachers can also learn a lot from their peers if given the opportunity. / It's also important to learn about the conditions that may affect attendance by youth rather than have biased assumptions. /Activities and projects are opportunities for all members (not just youth) to learn something new. |
| Digital technologies are important tools for learning and collaboration. | Digital applications can enhance collaboration while allowing for unique, individual work. |

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
