# Peer review of "The Reciprocal Power of Equitable, Intergenerational Learning: Exploring Perspectives of Undergraduate Students about Engaging in a University–Community Partnership Program"

_socsci, doi:10.3390/socsci12060349_

Round 1
Reviewer 1 Report
1-The nature of the study(qualitative) was clarified, but the nature of the methodology used and the type of research method adopted were not clarified, with justification for that in terms of its adoption in the study.
2-The objectives of the study were not presented.
3-The importance of the study was not raised.
4-The study variables were not determined procedurally.
5- New Leaf program is not sufficiently explained.
6- The "interview protocol" was not clarified and how it was built, described as a measurement tool, and the rationing process(honesty and reliability).
7- Why was only(11) students selected for a qualitative study like this? Wasn't it better to conduct a qualitative study of an exploratory nature using a questionnaire that includes open questions, and is distributed electronically to a large segment of students, to give a more comprehensive picture of their perceptions of the effectiveness of community participation for students undergraduates?
8- The study presented important results, which could be proposed hypotheses for future studies, which must be declared in the conclusion of the study, or presented as proposals that benefit researchers in expanding research on the subject.
9- It is also preferable to make recommendations to research professors and university officials to expand the application of these programs across various universities and disciplines.
Author Response
COMMENT | OUR RESPONSE TO THE EDITOR |
the nature of the methodology used and the type of research method adopted were not clarified, with justification for that in terms of its adoption in the study. | Within the introduction on pp. 2-3, we further clarified the appropriateness of using an ethnographic approach, given the lack of previous scholarship within this area. |
The objectives of the study were not presented. | We explicitly state our objective for this study that can be found on p. 2 in the introduction. |
importance of the study was not raised | Paragraph highlighting the relevance of the study and its findings was added in the abstract (p.1), so that there is a more direct reference to the importance highlighted within the beginning of the introduction (p. 2). |
study variables were not determined procedurally. | Within the Analytic Framework (p. 10), we clarified that our purpose, given the ethnographic approach is not on prescribed variables, but a grounded approach to seeing phenomena from our participants' perspective. |
New Leaf program is not sufficiently explained | In order to provide a clearer explanation of New Leaf, we moved all program-related descriptions to the study context starting on p. 7. We added a couple of other descriptors, and are open to any additional requests from the editor or reviewers. |
"interview protocol" was not clarified and how it was built, described as a measurement tool, and the rationing process(honesty and reliability). | Witin the methodology section (P. 8), we further clarified our development of our interview guide, which is based on our ethnographic approach. |
Why was only(11) students selected for a qualitative study like this? Wasn't it better to conduct a qualitative study of an exploratory nature using a questionnaire that includes open questions, | We addressed our choice of methodology and interviewing both in the introduction (see pp. 2-3) and the methodology (p.7). We explain that our approach aligns the guidance of prominent scholars who strongly encourage initial phases of qualitative inquiry before developing broadly administered questionnaires. Such an approach has been noted by scholars in measurement and survey design as key for developing instruments with strong evidence of validity (reference Wilson 2005 and Willis 2005). We have added these references to the reference list. |
The study presented important results, which could be proposed hypotheses for future studies, which must be declared in the conclusion of the study, or presented as proposals that benefit researchers in expanding research on the subject. | On p. 21 with the section titled "implications for future studies" we add a statement about what other researchers may benefit from in terms of doing such ethnographic work. Also, at the end of the paper (in Coda, see p. 21), we talk about potential future studies. At this point, we are not yet ready to think about testable ideas (i.e., hypotheses). |
It is also preferable to make recommendations to research professors and university officials to expand the application of these programs across various universities and disciplines. | Within the section titled "implications for future studies" on p. 20), we added an explicit example of proposed next steps for future work. |
Reviewer 2 Report
This paper provides interesting and important empirical results regarding the experience of undergraduate students who take part in a university-community partnership program. The authors provide information that contextualizes the partnership program, including its critical approach and goals, as well as providing rationale for studying the experiences of undergraduate students. The empirical results pulled from the interview transcripts are interesting an provide compelling support for involving undergraduate students in similar types of programming.
If the authors are looking for opportunities to improve the manuscript I would suggest that these opportunities are primarily related to organizational structure, including potentially reducing the number of arguments being made, and clarity regarding language used and the concepts that are employed. Below I provide some more specific suggestions for consideration by the author/s.
(1) Consistent use of terms throughout the paper specifically when describing the main group of study participants - undergraduate students. At times this group seems to also be called "higher education students" (abstract), "educators" (top of pg. 2), "university members" (middle of pg 2), "educator, activity facilitator, mentor" (bottom of pg. 3). Because there are so many different groups of "students" and "adults" involved in the project (e.g., the younger grade school participants/students, graduate students, faculty members) using these terms interchangeably with undergraduate student can become confusing. Is there an opportunity to simply use undergraduate students, or use this as a descriptor such as "undergraduate educators". There is also a mention of "community voice" on pg. 17 but it isn't entirely clear which community is being referred to.
(2) Ideas and concepts that are introduced in the paper are not represented in the abstract, and vice-versa. For example, it seems that in the body of the paper "engagement" and "silence" are two key concepts that are employed in the body of the paper, but are not clearly noted in the abstract. Additionally the seven key findings from the interviews are not noted in the abstract. Conversely, the abstract talks about translanguaging and a few other ideas that I don't see discussed in the paper. Is there an opportunity to more closely align abstract with key ideas in the paper?
(3) More clearly define core concepts. Once the key ideas and concepts are clarified it would be assistive to the reader if these were explicitly defined near the start of the paper. For example, while I am reading I get a sense of what "engagement" may mean from the literature but I'm not sure it is explicitly defined for the purposes of this specific paper.
(4) Possibility to reduce the number of arguments in order to make the paper more concise and increase clarity. I would encourage the author/s to read through the manuscript for descriptions or concepts that are not necessary to make the specific arguments. For example, it struck me as a reader that the parts of the paper that are related to digital technology included prior to discussing the interview results were very interesting but didn't quite have enough explanation for me to fully understand their relevance. The author/s may want to consider removing it and maybe even considering whether it could be a paper on it's own! Note: if this is kept in then elements of this aspect of the paper would need to be expanded. Specifically I would point to the need to include further explanation of the images of the New Leaf plant diaries that are on pg. 5.
Thanks for giving me the opportunity to review an interesting paper that is describing an exciting and valuable university-community partnership. Clearly this work has important implications for undergraduate education and the engagement of undergraduate students.
Author Response
COMMENT | OUR RESPONSE TO THE EDITOR |
Consistent use of terms throughout the paper specifically when describing the main group of study participants - undergraduate students. At times this group seems to also be called "higher education students" (abstract), "educators" (top of pg. 2), "university members" (middle of pg 2), "educator, activity facilitator, mentor" (bottom of pg. 3). Because there are so many different groups of "students" and "adults" involved in the project (e.g., the younger grade school participants/students, graduate students, faculty members) using these terms interchangeably with undergraduate student can become confusing. Is there an opportunity to simply use undergraduate students, or use this as a descriptor such as "undergraduate educators". There is also a mention of "community voice" on pg. 17 but it isn't entirely clear which community is being referred to. | The use of terminology that refers to undergraduate students was made more clear and consistent across the paper, including the abstract. |
Ideas and concepts that are introduced in the paper are not represented in the abstract, and vice-versa. For example, it seems that in the body of the paper "engagement" and "silence" are two key concepts that are employed in the body of the paper, but are not clearly noted in the abstract. Additionally the seven key findings from the interviews are not noted in the abstract. Conversely, the abstract talks about translanguaging and a few other ideas that I don't see discussed in the paper. Is there an opportunity to more closely align abstract with key ideas in the paper? | We added brief notes about engagement and silence in the abstract. Further, we believe that the revisions we have made that are delineated in this response help build greater cohesion between the abstract and the manuscript while also adhering to the word limit guideline for the abstract. If there are any specific requests, we are open to addressing them. |
More clearly define core concepts. Once the key ideas and concepts are clarified it would be assistive to the reader if these were explicitly defined near the start of the paper. For example, while I am reading I get a sense of what "engagement" may mean from the literature but I'm not sure it is explicitly defined for the purposes of this specific paper. | Within the introduction (p. 4), we referred to our choice of ethnography as a first step towards clarifying relevant core concepts, which may be useful in future studies. |
Possibility to reduce the number of arguments in order to make the paper more concise and increase clarity. I would encourage the author/s to read through the manuscript for descriptions or concepts that are not necessary to make the specific arguments. For example, it struck me as a reader that the parts of the paper that are related to digital technology included prior to discussing the interview results were very interesting but didn't quite have enough explanation for me to fully understand their relevance. The author/s may want to consider removing it and maybe even considering whether it could be a paper on it's own! Note: if this is kept in then elements of this aspect of the paper would need to be expanded. Specifically I would point to the need to include further explanation of the images of the New Leaf plant diaries that are on pg. 5. | Description of digital activities used in the new leaf program were moved to the context section of the methodology (p.7-8) as a way of addressing two reviewer comments, this (reducing information in the introduction) and the request for more information about the program. If there are specific requests to remove any other kinds of information from the introduction, we welcome the suggestions. |
Round 2
Reviewer 1 Report
The manuscript has been improved in many of its elements, which qualifies it for publication as such